# Microstructural Assessment of a Multiple-Intermetallic-Strengthened Aluminum Alloy Produced from Gas-Atomized Powder by Hot Extrusion and Friction Extrusion

**DOI:** 10.3390/ma13235333

**Published:** 2020-11-25

**Authors:** Tianhao Wang, Bharat Gwalani, Joshua Silverstein, Jens Darsell, Saumyadeep Jana, Timothy Roosendaal, Angel Ortiz, Wayne Daye, Tom Pelletiers, Scott Whalen

**Affiliations:** 1Pacific Northwest National Laboratory, Department of Energy, 902 Battelle Blvd., Richland, WA 99354, USA; tianhao.wang@pnnl.gov (T.W.); Bharat.Gwalani@pnnl.gov (B.G.); joshua.silverstein@pnnl.gov (J.S.); Jens.Darsell@pnnl.gov (J.D.); Saumyadeep.Jana@pnnl.gov (S.J.); Timothy.Roosendaal@pnnl.gov (T.R.); angel.ortiz@pnnl.gov (A.O.); 2Kymera International, 2601 Weck Drive, Research Triangle Park, NC 27709, USA; wayne.daye@kymerainternational.com (W.D.); tom.pelletiers@kymerainternational.com (T.P.)

**Keywords:** powder metallurgy, high-temperature aluminum alloy, transition metals, friction extrusion, microstructural evolution, calculation of phase diagrams

## Abstract

An aluminum (Al) matrix with various transition metal (TM) additions is an effective alloying approach for developing high-specific-strength materials for use at elevated temperatures. Conventional fabrication processes such as casting or fusion-related methods are not capable of producing Al–TM alloys in bulk form. Solid phase processing techniques, such as extrusion, have been shown to maintain the microstructure of Al–TM alloys. In this study, extrusions are fabricated from gas-atomized aluminum powders (≈100–400 µm) that contain 12.4 wt % TM additives and an Al-based matrix reinforced by various Al–Fe–Cr–Ti intermetallic compounds (IMCs). Two different extrusion techniques, conventional hot extrusion and friction extrusion, are compared using fabricating rods. During extrusion, the strengthening IMC phases were extensively refined as a result of severe plastic deformation. Furthermore, the quasicrystal approximant IMC phase (70.4 wt % Al, 20.4 wt % Fe, 8.7 wt % Cr, 0.6 wt % Ti) observed in the powder precursor is replaced by new IMC phases such as Al_3.2_Fe and Al_45_Cr_7_-type IMCs. The Al_3_Ti-type IMC phase is partially dissolved into the Al matrix during extrusion. The combination of linear and rotational shear in the friction extrusion process caused severe deformation in the powders, which allowed for a higher extrusion ratio, eliminated linear voids, and resulted in higher ductility while maintaining strength comparable to that resulting from hot extrusion. Results from equilibrium thermodynamic calculations show that the strengthening IMC phases are stable at elevated temperatures (up to ≈ 600 °C), thus enhancing the high-temperature strength of the extrudates.

## 1. Introduction

Aluminum alloys are widely used in numerous industries because of their excellent properties, including high specific strength, corrosion resistance, and formability [1]. In particular, the aerospace and automotive transportation sectors extensively depend on these alloys because of their high strength-to-weight ratio [2,3]. Among age-hardenable aluminum alloys, Al–Cu (2XXX series) and Al–Zn–Cu (7XXX series) alloys possess a higher strength than that of the more common Al–Si (6XXX series) alloys at ambient temperature. However, the strength of these aluminum alloys degrades significantly with increasing temperature; therefore, their applications are typically limited to 150–200 °C [4]. This is because their strengthening mechanisms are unstable at elevated temperatures, namely (1) coarsening of the Al matrix grains and (2) coarsening or dissolution of second-phase precipitates [5]. As a result, aluminum alloys capable of higher-temperature service enabled by more stable strengthening phases are desirable.

A recent study demonstrated a new approach for stabilizing metastable θ′ (Al_2_Cu) precipitates via microalloying with Mn and Zr. Mn and Zr atoms segregate to the matrix–θ′ interfaces and prevent θ′ precipitates from coarsening at elevated temperatures [6]. Sc can also effectively strengthen Al alloys at elevated temperatures because Al_3_Sc (L1_2_) precipitates have a very low lattice mismatch (1.34% at room temperature) with the aluminum matrix [7]. The lattice mismatch between the matrix and precipitates is critical for maintaining mechanical properties at elevated temperatures, and Al_3_Sc exhibits a low coarsening rate at temperatures up to 350 °C and remains coherent to the matrix [8]. Furthermore, the high-temperature stability of Al_3_Sc precipitates leads to a stabilization of the grain structure, which maintains grain boundary strengthening [9] and prevents grain growth via Zener drag [10]. However, Sc is very expensive, which limits industrial application. Another well-accepted pathway for improved thermal stability is the use of fine-scale, uniformly distributed semi-coherent and incoherent second phases. For example, the transition metals (TMs) Fe, Ti, Cr, V, Mo, and Zr exhibit limited solid solubility and low diffusivity into the aluminum matrix [11], which provides more thermally stable second phases with reduced Ostwald ripening [12]. For instance, Allied Signal Inc. developed a series of aluminum alloys (Al–Fe–Si–V) that remain stable at temperatures up to 400 °C. AA8009 alloy (Al-8.5Fe-1.7Si-1.3V wt %) is one of the commercially available alloys produced by rapid solidification (RS) along with a formation of thermally stable Al_12_(Fe,V)_3_Si (silicide) dispersoids [13]. The RS process of Al–TM alloys facilitates the formation of amorphous and quasi-crystalline (icosahedral) phases that simultaneously increase strength and ductility [14,15]. Apart from RS techniques, powder metallurgy consolidation techniques, including mechanical alloying (MA) [16], hot extrusion [17], and friction extrusion [18], have also been used for producing Al–TM alloys. For the reader’s convenience, Al–TM alloys with different chemical compositions and fabrication methods are tabulated in Table 1. Note that Al–TM alloys are not typically made by conventional casting because the formation of continuous intermetallic compound (IMC) phases is detrimental to mechanical properties. During severe plastic deformation of Al–TM powders, a microstructure with ultrafine grains and second phases is shown to enhance mechanical properties [16,17,18]. Compared to RS, powder metallurgy consolidation techniques are more economical, industrially viable, and faster at producing Al–TM alloys for engineering applications.

Despite the body of research on rapidly solidified Al–TM alloys, studies comparing the microstructure for a given alloy extruded by different methods are lacking. In this study, two extrusion methods—conventional hot extrusion and friction extrusion—were applied to produce rods from Al–12.4TM powders (Al with 12.4 wt % TMs). Friction extrusion, also known as shear-assisted processing and extrusion (ShAPE), has been applied for fabricating AZ91 magnesium tubes [22] and rods [23] and ZK60 magnesium tubes [24,25]. The mechanism of friction extrusion is different from that of conventional extrusion because in addition to the linear extrusion force, a rotational shear force is simultaneously applied via a spinning die that imparts extensive deformation. In this study, the microstructure evolution of Al–12.4TM under conventional hot extrusion and friction extrusion at different extrusion ratios (ERs) was investigated. In addition, mechanical properties are compared with common aluminum alloys at ambient and elevated temperatures. 

## 2. Materials and Methods 

### 2.1. Base Materials

Feedstock powder was provided by SCM Metal Products, Inc. (Durham, NC, USA), a division of Kymera International. The powder used in this study, designated as Al–12.4TM (Al with 12.4 wt % TMs, including Ti, Cr, Mn, Mo, Fe, and Si, and other trace additives), was produced by induction melting followed by inert gas atomization and screening to particle sizes of less than 400 µm. Powders were stored in sealed containers in an ambient environment, i.e., powders were not stored in an inert atmosphere. 

### 2.2. Friction Extrusion and Hot Extrusion Processes

Friction extrusion was performed using a Transformation Technology, Inc. (Elkhart, IN, USA). (TTI) LS2-2.5 friction stir welder capable of 130 kN axial force while rotating at 1950 rpm with 570 Nm of torque. The tooling was composed of an extrusion die and powder container that has been fully described previously [18]. In general, the extrusion die was fabricated from H13 tool steel with an outer diameter of 31.62 mm and an inner diameter of 5.0 mm that defined the outer diameter of the extrusion. Spiral grooves were machined into the die face to facilitate material flow into the extrusion orifice. Temperature was sensed using a thermocouple embedded 1 mm from the die face. The powder container was machined from H13 tool steel to an inner diameter of 31.75 mm, which produced an ER of 40.3. The rotating die was plunged at 3.81 mm/min, and the die rotation was slowly reduced from 500 to 300 rpm to stabilize the temperature at 450 °C. Hot-extruded specimens were provided by Kymera International, and the processing details are described in detail elsewhere [17]. In general, ERs ranged from 6 to 25, and the extrudate exit temperature ranged from 525 to 550 °C. Schematics for both friction extrusion and hot extrusion are displayed in Figure 1a,b, respectively.

### 2.3. Sample Preparation and Characterization Methods

Both friction extrusion and hot extrusion specimens were cut along the longitudinal section using a diamond saw (Figure 2). Specimens for microstructural analyses were mounted in epoxy and polished to a final surface finish of 0.05 μm using colloidal silica. Optical microscopy (OM) was performed using an Olympus BX-51 fluorescence motorized microscope (Olympus, Tokyo, Japan). Scanning electron microscopy (SEM) analyses were performed using a JEOL 7600F field emission scanning electron microscope (JEOL, Tokyo, Japan) with energy-dispersive X-ray spectroscopy (EDS) using an accelerating voltage of 20 keV and a working distance of 10–15 mm. X-ray diffraction (XRD) (Bruker AXS Inc., Madison, WI, USA) was conducted on Al–12.4TM powders and Al–12.4TM extrudates to evaluate the evolution of second phases within the Al matrix. Tensile testing on extrudates at ambient and elevated temperatures was conducted according to ASTM E8 and ASTM E21, respectively, for friction-extruded and hot-extruded specimens.

### 2.4. Equilibrium Thermodynamic Calculation

The experimentally observed phases obtained from the different extrusion methods are compared with the equilibrium phases calculated using the computational thermodynamic (CALculation of PHAse Diagrams, CALPHAD) approach. This comparison also takes advantage of input from the TCHEA3 database using Thermo-Calc software (TCHEA3, Thermo-Calc Software, Stockholm, Sweden).

## 3. Results and Discussion

### 3.1. Microstructural Investigation 

OM images were taken from extrudates obtained via hot extrusion and friction extrusion to study the microstructural differences resulting from the two methods at different ERs (Figure 3). Extrudates made via hot extrusion with ER = 6 had inhomogeneous microstructures with narrow stringer-like features that averaged 50 µm wide and were highly elongated in the extrusion direction. These stringers form as powder particles and co-deform while being sheared in the extrusion direction. A detailed SEM analysis of stringer-like features is presented in the next section. Similar microstructural features are observed in hot-extruded specimens with ER = 25; however, the stringer-like features were observed to be much finer than that with ER = 6 (Figure 3(b1–b3)), with an average width of 5 µm. In addition, stringer-like features at the extrudate center (Location 1, as shown in Figure 3(a1)) are coarser than those at the extrudate periphery (Location 3, as shown in Figure 3(a3)) for ER = 6. This suggests that shear deformation is more severe at the extrudate periphery than at the center, as is common during extrusion. The resolution of OM is not high enough to reveal microstructural variances at ER = 25, which suggests that as expected, deformation becomes more severe as ER increases (e.g., roughly 4 times from ER = 6 to ER = 25). Although increasing by a factor of only 1.6, the microstructure of friction extrusion with ER = 40 is dramatically different compared to hot extrusion with ER = 25. For friction extrusion, the microstructure is also observed to be inhomogeneous but with a very different morphology than that which results from hot extrusion. Rather than the elongation of features in the extrusion direction, onion-ring-like features are observed at the center of friction-extruded material (Location 1, as shown in Figure 3(c1)). These features are created transverse to the extrusion direction by the combination of linear and rotational shear, which is a process that is intrinsic to friction extrusion, as the material enters the extrusion orifice. Similar features are commonly observed in friction stir welding and processing [26]. As shown in Figure 3(c2,c3), the microstructural features at locations that are radially outward from the centerline are more homogeneous.

A more detailed microstructural analysis of the powder precursor, hot extrudates, and friction extrudates was conducted via SEM (Figure 4). Since there is no significantly evident microstructural difference between the center and periphery region for hot extrusions, only the center regions of hot extrusions were further investigated via SEM (Figure 4(b1–b3,c1–c3)). For friction extrusions, evident microstructural differences were observed with OM; therefore, both the center and periphery regions of friction extrusions were investigated via SEM (Figure 4(d1–d3,e1–e3)). The secondary phases (IMC phases) present in the starting powder display wide ranges of size and morphology due to variations in cooling rate across the spread of particle sizes, with chemistry and phase variety extensively characterized in prior work [17,18]. In general, the size of the IMC phases in the starting powders are larger than that in extruded materials due to refinement that occurs during shear deformation. For hot extrusions made with ER = 6 (Figure 4(b1–b3)), the size of the IMC phase is not homogenous. Secondary phases within the stringer-like features (marked with yellow arrows in Figure 4(b1)) are only slightly refined compared to the starting powder, while adjacent regions (marked with red arrows in Figure 4(b1) and shown with higher magnification in Figure 3(b3)) are highly refined compared to the starting powder. As for hot extrusions made with ER = 25, stringer-like features are no longer evident, and the entire microstructure is highly refined compared to the starting powder, as shown in Figure 4(c1–c3). For friction extrusion made with ER = 40, regions of lightly refined IMCs are observed as 5–25 µm thick bands that are oriented transverse to the extrusion direction and are labeled with yellow arrows in Figure 4(d1). Material adjacent to the transverse bands is observed to be highly refined compared to the starting powder. The secondary phases of friction-extruded material at Location 3 (Figure 4(e1–e3)) are highly refined compared to those at Location 1, which is along the centerline (as shown in Figure 4(d1–d3)). The thickness of the highly refined skin along the periphery of the extrusion has a thickness of approximately 500 µm. Areas of secondary phases in the periphery region of friction extrusions tend to be aligned with the extrusion direction, as shown in Figure 4(e2,e3). In prior work [18], it was found that friction-extruded Al–12.4TM powder had twice the ductility of hot-extruded material with ER = 6 and ER = 25 without a reduction in yield or ultimate strength. Using information obtained in the present analysis, we hypothesize that the highly refined skin developed during friction extrusion is effective at reducing stress concentrations that lead to an initiation of surface cracks, thereby improving elongation without sacrificing strength. At the center of a friction-extruded area, material primarily flows transverse to the extrusion direction; however, in the periphery region, material flow parallel to the extrusion direction is dominant. Some voids observed in hot extrusions are aligned parallel to the extrusion direction (labeled by red arrows in Figure 4(b1,c1)). Note that there is no evidence of voiding in the friction extrusions. This is likely due to the additional mixing provided by the rotational component of friction extrusion that is not present in hot extrusion. Voids vanishing in friction extrusions can be another factor of improved elongation.

To investigate the chemical composition distribution of the various secondary phases in the matrix, the staring powders and extrusions were examined used EDS. Figure 5(a1–a5) displays results for the starting Al–12.4TM powders that indicate two distinct IMC morphologies: equiaxed faceted particles and conjoined rectangular structures. Our previous transmission electron microscopy work [18] showed that the equiaxed faceted IMC phase that is rich in Fe and Cr and lean in Ti is a quasicrystal approximant (70.4 wt % Al, 20.4 wt % Fe, 8.7 wt % Cr, 0.6 wt % Ti) and that the conjoined rectangular IMC phase that is rich in Ti and Cr and lean in Fe is an Al_3_Ti-type structure (66.4 wt % Al, 28.8 wt % Ti, 4.7 wt % Cr). IMC phases comminuted during the processing of hot extrusion and friction extrusion, as shown in Figure 5(b1–b5,c1–c5,d1–d5,e1–e5). Fe and Cr tend to separate from each in smaller-scale features, forming one IMC phase that is rich in Fe and another IMC phase that is rich in Cr, as shown in Figure 5(c1–c5,e1–e5). Our previous work [18] suggested that the newly formed IMCs rich in Fe are based on an Al_3.2_Fe-type structure (65 wt % Al, 35 wt % Fe) and that the newly formed IMC rich in Cr could be the non-stoichiometric Al_45_Cr_7_-type structure (78 wt % Al, 11 wt % Cr, 9 wt % Fe, 2 wt % Ti). The phase separation tendencies of Fe and Cr in Fe–Cr, Fe–Cr–Al, and Fe–Cr–Al–Ti alloys at elevated temperatures [27,28,29] or under irradiation [30,31] have been previously reported, resulting in similar elemental separation. IMC phases with larger sizes (several microns) remain rich in both Fe and Cr (Figure 5(c1–c5,d1–d5)) while maintaining compositions similar to the starting powders. Al_3_Ti-type phases present in the powder precursor were fractured into finer particles during processing and, as a result of the nanoscale size of Ti (as shown in Figure 5(a5,b5,c5,d5,e5)), will likely dissolve into and perhaps precipitate out of the Al matrix during hot extrusion and friction extrusion. Furthermore, higher magnification SEM and EDS analyses of a selected region located within the friction extrusion center region (Figure 6) suggest that the initial stage of the phase transformation process occurs within the Al–Fe–Cr alloy system during friction extrusion. Figure 6 shows that the Fe-rich phase (labeled by black arrows) forms around the pre-existing quasicrystal approximant phase (70.4 wt % Al, 20.4 wt % Fe, 8.7 wt % Cr, 0.6 wt % Ti). By using the following thermal–mechanical cycle in the current alloy system, the quasicrystal approximant phases were further broken down along with phase transformation as verified in the periphery region of the friction extrusion (as shown in Figure 5(e1–e4)). Note that the mechanical effect in the thermal–mechanical cycle is more evident for the periphery region than for the center region of the friction extrusion.

The distinct crystal structures of the different IMC phases within starting powders and friction-extruded material were investigated by XRD analysis (Figure 7a–d). It was observed that the Al (Face Centered Cubic, FCC) phase dominates and that hexagonal Al_3.2_Fe, monoclinic Al_45_Cr_7_, and BCT Al_3_Ti are unambiguously indexed for friction extrusion. Note that the quasicrystal approximant phases (70.4 wt % Al, 20.4 wt % Fe, 8.7 wt % Cr, 0.6 wt % Ti) observed in starting powders are indexed as Al_3.2_Fe because they have similar lattice parameters. These observations are consistent with those reported earlier by Whalen et al. using transmission electron microscopy [18]. 

The equilibrium structure of the Al–12.4TM alloy is assessed using the Thermo-Calc TCHEA3 database. The equilibrium phase fraction in this alloy as a function of temperature is calculated and shown in Figure 8. The various IMCs within the alloy exhibit stability at temperatures up to ≈ 650 °C, beyond which the onset of liquation is observed. At room temperature, the molar fractions of different intermetallic phase structures calculated from equilibrium solution thermodynamics calculations are 50% Al_13_Fe_4_, 37% Al_13_Cr_2_, 11% FCC/L1_2_, and 2% BCT/DO_22_. Other intermetallic phases, such as Al_4_Cr and Al_11_Cr_2_, are predicted to stabilize at high temperature of 600 °C. The Al_13_Fe_4_ phase predicted by CALPHAD is stoichiometrically close to the experimentally observed Al_3.2_Fe phase, and Al_13_Cr_2_ is stoichiometrically close to the Al_45_Cr_7_ phase that was experimentally identified in this work. A DO_22_ (Al_3_Ti) phase is observed both theoretically as well as experimentally. Interestingly, the equilibrium fraction of the Al (FCC) phase is predicted to be low, which is inconsistent with our experimental findings. The experimental results show that the matrix Al (FCC) phase forms the major phase of the microstructure, which could be a result of the slow kinetics growth and coarsening of the multiple intermetallic phases. The thermodynamic calculations predict the type of equilibrium phases; however, our current extrusion processes do not result in an equilibrium phase because of the severe straining and insufficient annealing times required to reach equilibrium. Moreover, these compositionally conjugate intermetallic phases can interpenetrate with each other and therefore reduce the kinetics for coarsening. 

### 3.2. Mechanical Response Characterization 

To compare the mechanical responses of various aluminum alloys at different temperatures, an Ashby plot was constructed to compare the ultimate tensile strength of commercially available Al–12.4TM as a function of temperature with other commercially available aluminum alloys (Figure 9). Note that the temperature at which the strength starts to fall sharply is defined as the maximum service temperature. Figure 9 shows that the maximum service temperature of most aluminum alloys is less than ≈200 °C except for the AA8009 alloy, which is a rapidly solidified Al–TM alloy with a chemical composition of Al–Fe–Si–V. At a service temperature of 250–300 °C, the only commercially available alloy with a higher tensile strength than Al–12.4TM that the authors are aware of is AA8009 (labeled in Figure 9). 

The tensile strengths and elongations at elevated temperatures of various Al–TM alloys produced by rapid solidification are compared with friction extrusion and hot extrusion in Figure 10. As shown in Figure 10, the tensile strength of the Al–12.4TM alloy fabricated via hot and friction extrusions is comparable to other Al–TM alloys made via rapid solidification methods. Furthermore, ambient temperature elongation of an Al–12.4TM alloy fabricated through friction extrusion is enhanced significantly compared to that of Al–TM alloys produced through rapid solidification or hot extrusion methods (Figure 11). Note that the elongation of rapidly solidified Al–TM alloys increases with temperature, and surprisingly, the elongation of Al–12.4TM fabricated via friction extrusion decreased as temperature increased. The mechanism of ductility loss at elevated temperature of friction extruded Al-12.4TM alloy is unclear and is currently being investigated.

## 4. Conclusions

For hot extrusion, deformation at the extrusion periphery region is more severe than at the extrusion center region. While for friction extrusion, material flow perpendicular to the extrusion direction is dominant at the center region, and material flow parallel to the extrusion direction is dominant at the extrusion periphery region.Friction-extruded Al–12.4TM has a comparable strength (>200 MPa at 300 °C) but a higher ductility (>15% at ambient temperature) than hot-extruded Al–12.4TM. This is likely due to the elimination of voiding and finer second phases, which is enabled by the additional high-shear component introduced during friction extrusion.During extrusion, the quasicrystal approximant IMC phase (70.4 wt % Al, 20.4 wt % Fe, 8.7 wt % Cr, 0.6 wt % Ti) was broken into small pieces. Fe and Cr tend to separate from each other, forming new IMC phases of the Al_3.2_Fe–type (65 wt % Al, 35 wt % Fe) and non-stoichiometric Al_45_Cr_7_-type structures (78 wt % Al, 11 wt % Cr, 9 wt % Fe, 2 wt % Ti).The rectangular IMC phases (66.4 wt % Al, 28.8 wt % Ti, 4.7 wt % Cr) with Al_3_Ti-type structures in the powder precursor break into small pieces and are dissolved into the Al matrix during hot extrusion and friction extrusion processes.A fine-scale distribution of multiple intermetallic phases is achieved in this material using friction extrusion. The competition of coarsening at the interfaced between these precipitates pins them and restricts their extent of coarsening even at high temperature.

## Figures and Tables

**Figure 1 materials-13-05333-f001:**
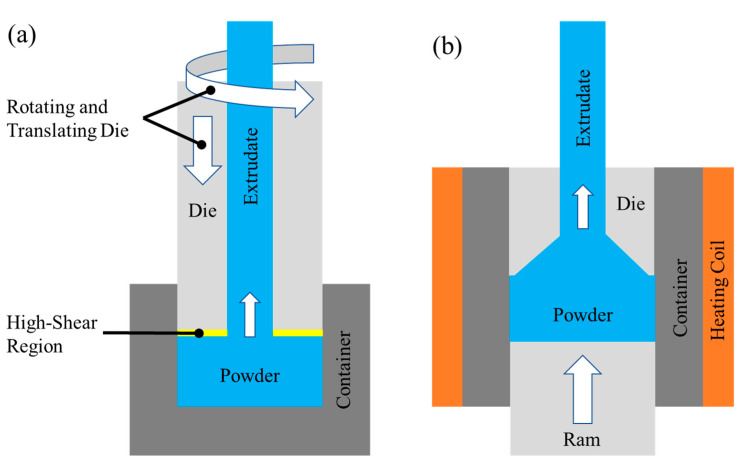
Schematic of (**a**) friction extrusion and (**b**) hot extrusion on consolidated powders.

**Figure 2 materials-13-05333-f002:**
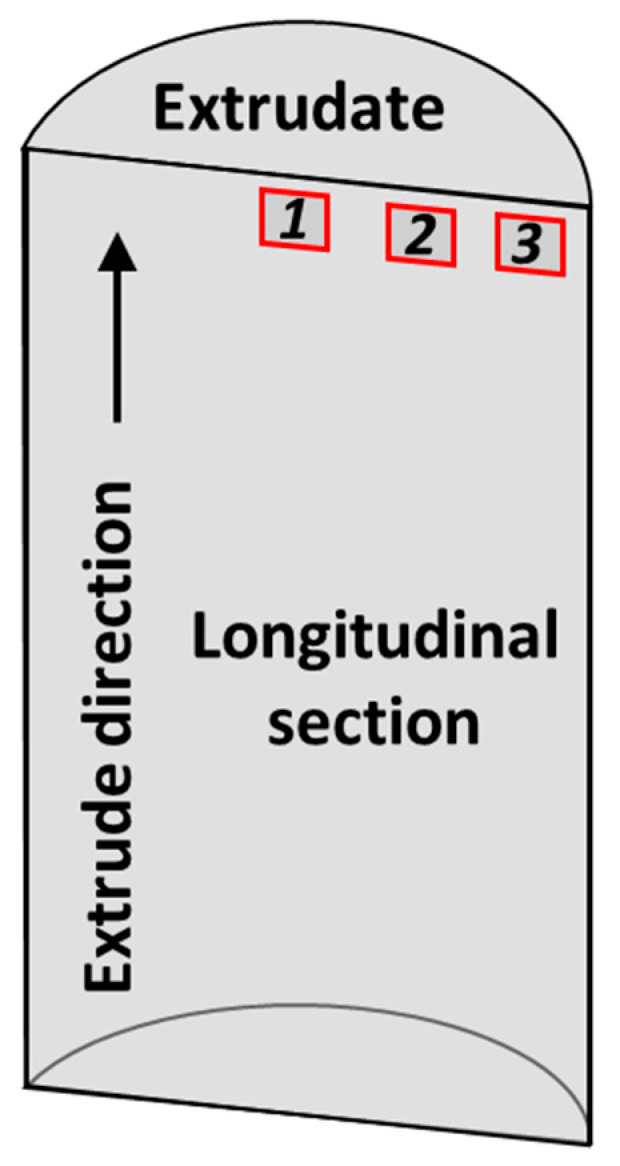
Location of samples extracted from extrudates for microstructural characterization.

**Figure 3 materials-13-05333-f003:**
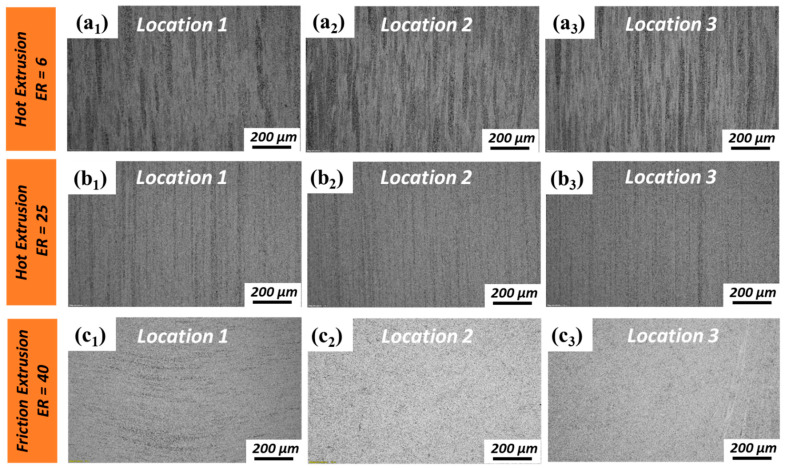
Optical microscopy (OM) images of the longitudinal sections of specimens made by hot extrusion and friction extrusion: (**a1**–**a3**) hot extrusion made with extrusion ratio (ER) = 6; (**b1**–**b3**) hot extrusion made with ER = 25; and (**c1**–**c3**) friction extrusion made with ER = 40.

**Figure 4 materials-13-05333-f004:**
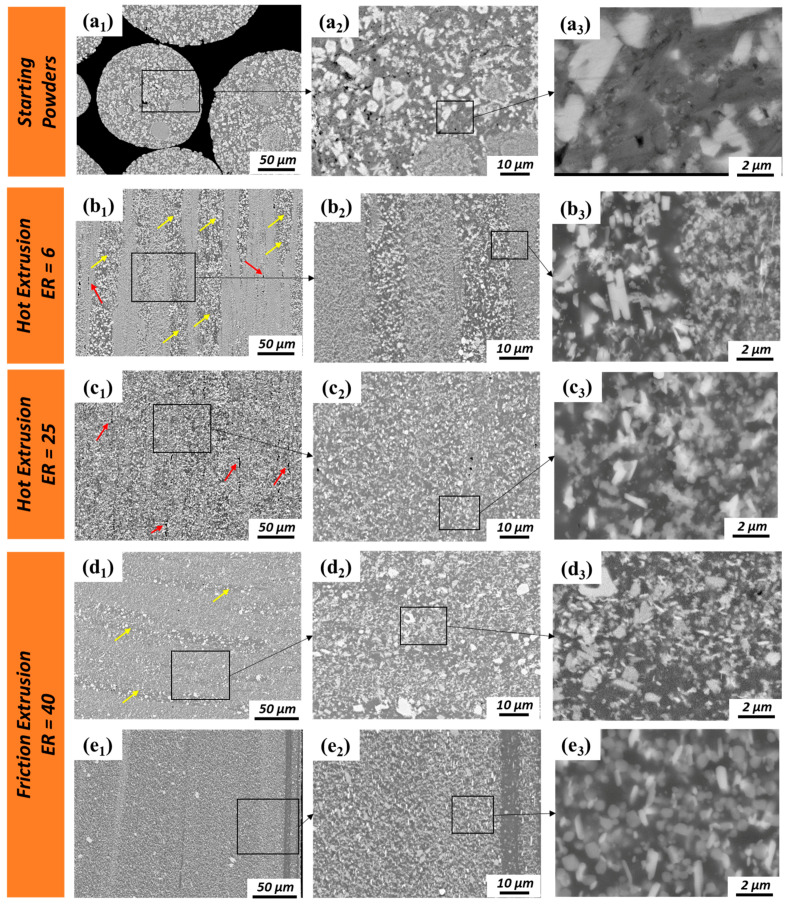
SEM images of (**a1**–**a3**) Al–12.4TM starting powders and longitudinal sections of extrudates; (**b1**–**b3**) Location 1 of hot extrusion with ER = 6; (**c1**–**c3**) Location 1 of hot extrusion with ER = 25; (**d1**–**d3**) Location 1 and (**e1**–**e3**) Location 3 of friction extrusion with ER = 40. Note that the stringer and transverse onion-ring regions with large secondary phases are labeled with yellow arrows. Linear voids are labeled with red arrows. TM: transition metal.

**Figure 5 materials-13-05333-f005:**
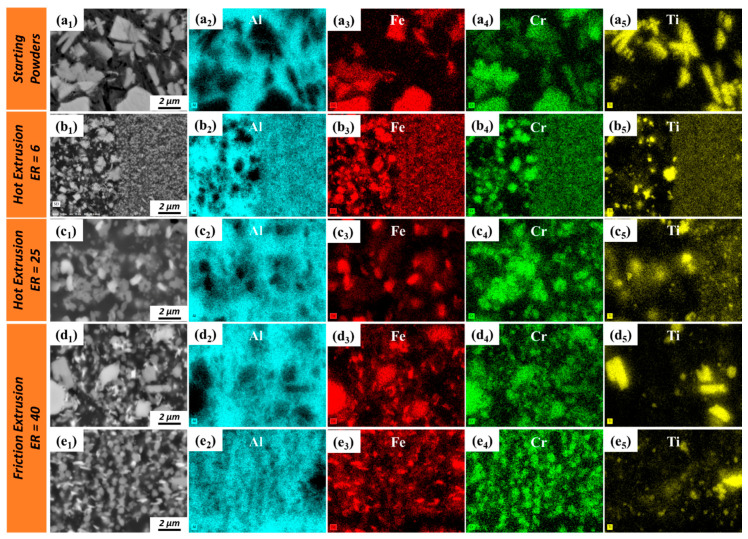
Energy-dispersive X-ray spectroscopy (EDS) analysis of (**a1**–**a5**) Al–12.4TM starting powders and the longitudinal section of extrudates; (**b1**–**b5**) Location 1 of hot extrusion with ER = 6; (**c1**–**c5**) Location 1 of hot extrusion with ER = 25; (**d1**–**d5**) Location 1 and (**e1**–**e5**) Location 3 of friction extrusion with ER = 40.

**Figure 6 materials-13-05333-f006:**
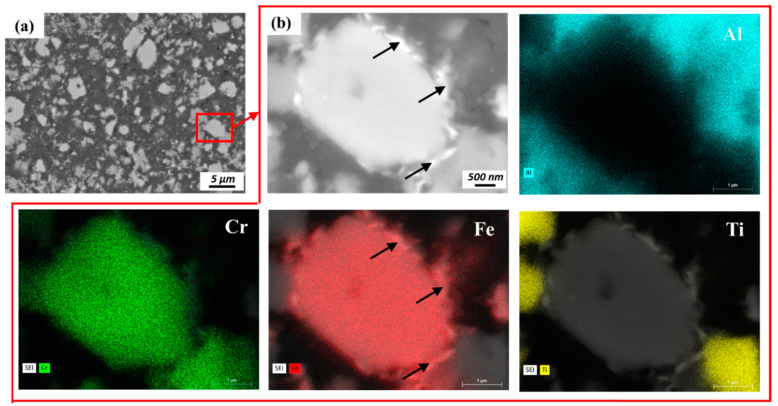
(**a**) SEM and (**b**) EDS analyses reveal the phase separation of the quasicrystal approximant (70.4 wt % Al, 20.4 wt % Fe, 8.7 wt % Cr, 0.6 wt % Ti) into Fe-rich and Cr-rich phases.

**Figure 7 materials-13-05333-f007:**
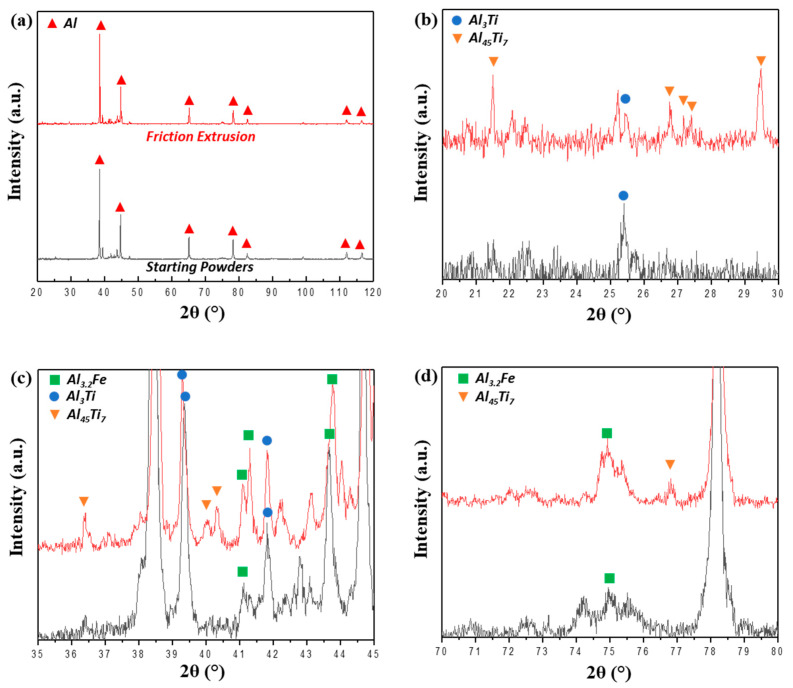
XRD analysis of Al–12.4TM powders and friction extrusions with different ranges of 2θ: (**a**) 20°–120°, (**b**) 20°–30°, (**c**) 35°–45°, and (**d**) 70°–80°.

**Figure 8 materials-13-05333-f008:**
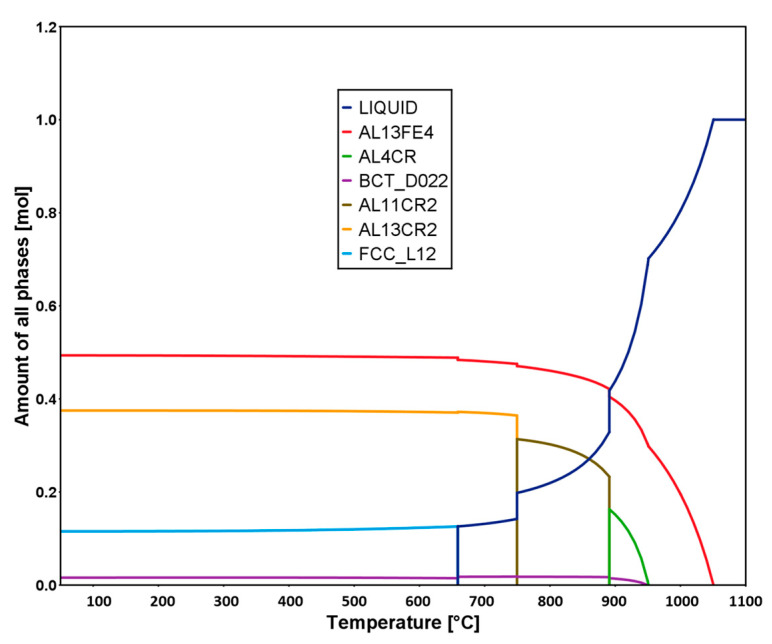
Calculated molar fraction of equilibrium phases as a function of the temperature for the Al–12.TM alloy used in this study.

**Figure 9 materials-13-05333-f009:**
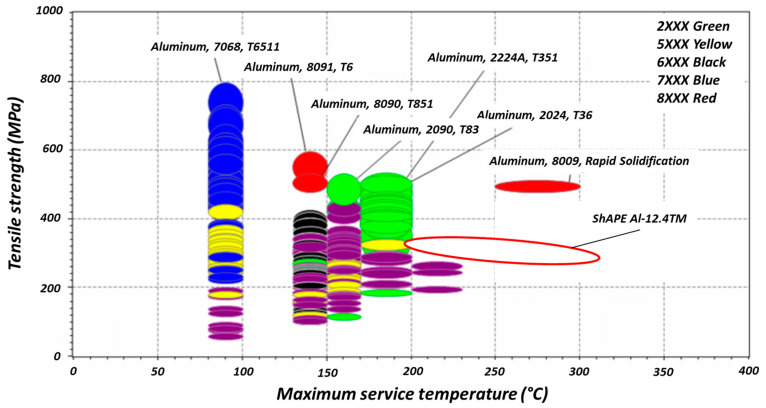
A comparison of the tensile strengths of various aluminum alloys at elevated temperatures.

**Figure 10 materials-13-05333-f010:**
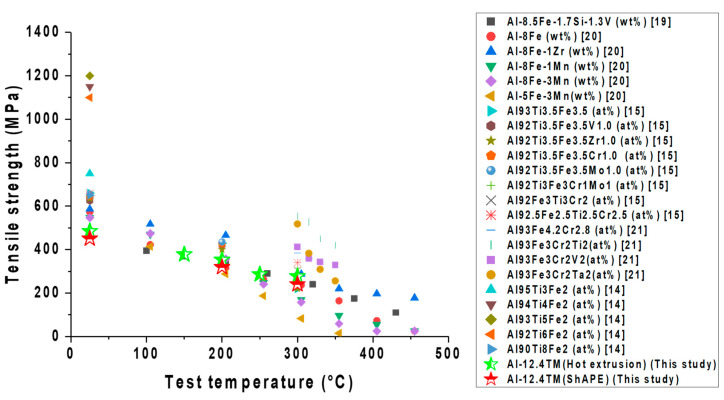
Comparison of the tensile strengths of various Al–TM alloys at elevated temperatures.

**Figure 11 materials-13-05333-f011:**
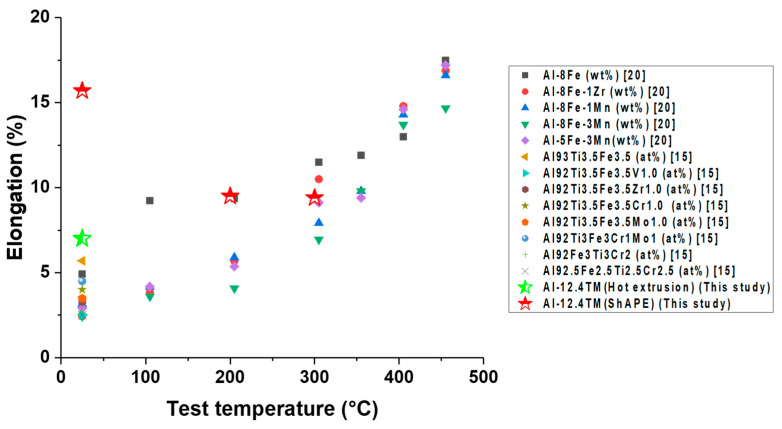
Comparison of elongation for various Al–TM alloys at elevated temperatures.

**Table 1 materials-13-05333-t001:** Chemical composition of various Al–TM alloys fabricated via different methods.

Elements	Al	Fe	Si	V	Zr	Cr	Mo	Mn	Ti	Nb	Ta	Methods	Refs.
Composition (atomic%)	Balance	8.5	1.7	1.3	–	–	–	–	–	–	–	RS	[19]
	Balance	8	–	–	–	–	–	–	–	–	–	RS	[20]
	Balance	8	–	–	1	–	–	–	–	–	–	RS	[20]
	Balance	8	–	–	–	–	–	1	–	–	–	RS	[20]
	Balance	8	–	–	–	–	–	3	–	–	–	RS	[20]
	Balance	5	–	–	–	–	–	3	–	–	–	RS	[20]
	Balance	3.5	–	–	–	–	–	–	3.5	–	–	RS	[15]
	Balance	3.5	–	1	–	–	–	–	3.5	–	–	RS	[15]
	Balance	3.5	–	–	1	–	–	–	3.5	–	–	RS	[15]
	Balance	3.5	–	–	–	1	–	–	3.5	–	–	RS	[15]
	Balance	3.5	–	–	–	–	1	–	3.5	–	–	RS	[15]
	Balance	3	–	–	–	1	1	–	3	–	–	RS	[15]
	Balance	3	–	–	–	2	–	–	3	–	–	RS	[15]
	Balance	2.5	–	–	–	2.5	–	–	2.5	–	–	RS	[15]
	Balance	4.2	–	–	–	2.8	–	–	–	–	–	RS	[21]
	Balance	3	–	–	–	2	–	–	2	–	–	RS	[21]
	Balance	3	–	2	–	2	–	–	–	–	–	RS	[21]
	Balance	3	–	–	–	2	–	–	–	2	–	RS	[21]
	Balance	2	–	–	–	2	–	–	–	–	2	RS	[21]
	Balance	2	–	–	–	–	–	–	3	–	–	RS	[14]
	Balance	2	–	–	–	–	–	–	4	–	–	RS	[14]
	Balance	2	–	–	–	–	–	–	5	–	–	RS	[14]
	Balance	2	–	–	–	–	–	–	6	–	–	RS	[14]
	Balance	2	–	–	–	–	–	–	8	–	–	RS	[14]
	Balance	3	–	–	–	2	–	–	2	–	–	MA	[16]

## Data Availability

The raw and processed data required to reproduce these findings cannot currently be shared as a result of technical limitations and developing intellectual property.

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
