# Peer review of "Microstructural Assessment of a Multiple-Intermetallic-Strengthened Aluminum Alloy Produced from Gas-Atomized Powder by Hot Extrusion and Friction Extrusion"

_materials, 2020, doi:10.3390/ma13235333_

Round 1

Reviewer 1 Report

This paper presents useful results of the fabrication of Al-12.4TM alloys via hot and friction extrusions. Discussion of the results is well-written and comprehensive. Reviewer's suggestions are given below.

  1. The source or reference of Figure 9 is required. 
  2. As shown in Figure 10, tensile strengths of Al-12.4TM alloys fabricated by this study at room temperature are relatively low. Authors are suggested to provide explanation for this result.
  3. In Figure 11, no elongation data of Al-12.4TM alloys obtained at elevated temperatures are provided. What are the reasons for missing those data?
  4. What is the effect of different extrusion ratios on mechanical properties (for example, tensile strength) of the product fabricated by friction extrusion? Did authors observe such influence in this study?

Author Response

This paper presents useful results of the fabrication of Al-12.4TM alloys via hot and friction extrusions. Discussion of the results is well-written and comprehensive. Reviewer's suggestions are given below.

  1. The source or reference of Figure 9 is required.

Response: The plot was constructed via Ashby Plotting Software. (Line 271)

  1. As shown in Figure 10, tensile strengths of Al-12.4TM alloys fabricated by this study at room temperature are relatively low. Authors are suggested to provide explanation for this result.

Response: The current Al-12.4 TM was fabricated through solid-phase friction extrusion, while other products displayed in Figure 10 was fabricated through rapid solidification. Rapid solidification of Al–TM alloys facilitates the formation of amorphous and quasi-crystalline phases that simultaneously increase strength and ductility. Compared to rapid solidification, powder metallurgy consolidation techniques (friction extrusion in this study) are more economical, industrially viable, and faster at producing Al–TM alloys for engineering applications.

  1. In Figure 11, no elongation data of Al-12.4TM alloys obtained at elevated temperatures are provided. What are the reasons for missing those data?

Response: Elongation data for 200 and 300 °C were added in the Figure 11. Additionally, the relevant explanation was added in the manuscript. (Lines 287-293)

  1. What is the effect of different extrusion ratios on mechanical properties (for example, tensile strength) of the product fabricated by friction extrusion? Did authors observe such influence in this study?

Response: By changing the extrusion ratio during friction extrusion, the material flow behavior and processing temperature changed significantly. And the diameter of friction extruded product varies with different extrusion ratios as well. It is challenging to compare them directly. What we observe so far: (i) Both microstructural homogeneity and processing temperature tend to increase with extrusion ratio increasing; (ii) The hardness value reduces with processing temperature increasing.

Reviewer 2 Report

The paper proposed here, aims at studying the microstructure and mechanical properties of Intermetallic Strengthened aluminum alloy produced by extrusion and friction extrusion.

I think the results are interesting but the mechanical properties part is a little weak. The authors give the evolution of the strength with temperature but not the evolution of the ductility which will be the key parameter since the strength is a little bit lower than the others alloys developed since now. Moreover, the tests have been carried out so they must be included.

There are also a few others comments:

. the composition is not explicitly given. The paper should include it

. the “ER” (page 3) should be defined in the text

. this a pity the ER is not the same between the two processes used. The comparison is then more complicated. How can we be sure that hot extrusion with ER=40 will not permit to get the same microstructure as friction extrusion ?

. the intermetallic phases are said to be quasicrystals because an other study proved it. However, how we be sure that this is the case here ?

. to compare the obtained phases with thermocalc calculation, I think the fraction of the phases found should be calculated from microscopy. Even if the matrix does not appear in the prediction, the respective fraction of the phases should be discussed.

. concerning the Ashby map, the maximum service temperature should be defined

. once again, the results concerning the evolution of the elongation with temperature must be included in order to fairly compare the alloy studied here with the literature

Author Response

The paper proposed here, aims at studying the microstructure and mechanical properties of Intermetallic Strengthened aluminum alloy produced by extrusion and friction extrusion.

I think the results are interesting but the mechanical properties part is a little weak. The authors give the evolution of the strength with temperature but not the evolution of the ductility which will be the key parameter since the strength is a little bit lower than the others alloys developed since now. Moreover, the tests have been carried out so they must be included.

There are also a few others comments:

  1. the composition is not explicitly given. The paper should include it.

Response: SCM Metal Products provided the powders “designated as Al–12.4TM (Al with 12.4 wt.% TMs, including Ti, Cr, Mn, Mo, Fe, and Si, and other trace additives) (Lines 89-90)” However, they cannot share the exact chemical composition due to commercial confidentiality.

  1. the “ER” (page 3) should be defined in the text

Response: Extrusion ratio (ER) has been added to line 84

  1. this a pity the ER is not the same between the two processes used. The comparison is then more complicated. How can we be sure that hot extrusion with ER=40 will not permit to get the same microstructure as friction extrusion?

Response: Authors completely agree with the comments. To achieve a higher extrusion ratio during hot extrusion, a higher machine force is required. The highest hot extrusion ratio was performed on what we believe to be the largest extrusion press on the east coast of the USA (8,600 ton press). With friction extrusion, the required machine force was relatively low. This is because the additional shear force on the powders during friction extrusion process can heat up and soften the powders before they get extruded.

  1. the intermetallic phases are said to be quasicrystals because another study proved it. However, how we be sure that this is the case here?

Response: Quasicrystal phases in the precursor powders should be constant with previous studies since we are using same precursor powders.

  1. to compare the obtained phases with thermocalc calculation, I think the fraction of the phases found should be calculated from microscopy. Even if the matrix does not appear in the prediction, the respective fraction of the phases should be discussed.

Response: The second phases existing in the matrix are very complicated, and size of those second phases are very small. Therefore, it is very challenging to quantify the phase fabrications via high-magnification images. The aim of current thermocalc calculation is to explain why quasicrystal approximate IMC (rich in both Fe and Cr) in precursors can be replaced by Al3.2Fe type and Al45Cr7 type structures during extrusion process.

  1. concerning the Ashby map, the maximum service temperature should be defined.

Response: Relevant content was added in the manuscript. (Lines 273-274)

  1. once again, the results concerning the evolution of the elongation with temperature must be included in order to fairly compare the alloy studied here with the literature.

Response: Elongation data for 200 and 300 °C were added in the Figure 11. And relevant content was added in the manuscript. (Lines 287-293)

Round 2

Reviewer 2 Report

I think the corrections made by the authors increased the quality of the paper. Even if some points have not been fully addressed, the paper could be published.